# Structure and Magnetic Properties of Mechanosynthesized Nanocrystalline Fe_2_CrSi Heusler Alloy

**DOI:** 10.3390/nano13233024

**Published:** 2023-11-26

**Authors:** Elżbieta Jartych, Paulina Jaskółowska, Dariusz Oleszak, Marek Pękała

**Affiliations:** 1Department of Electronics and Information Technology, Faculty of Electrical Engineering and Computer Science, Lublin University of Technology, 20-618 Lublin, Poland; 2Faculty of Materials Science and Engineering, Warsaw University of Technology, 02-507 Warsaw, Poland; paulina.jaskolowska.stud@pw.edu.pl (P.J.); dariusz.oleszak@pw.edu.pl (D.O.); 3Chemistry Department, Warsaw University, 02-089 Warsaw, Poland; pekala@chem.uw.edu.pl

**Keywords:** Heusler alloy, mechanical alloying, Mössbauer spectroscopy, magnetization

## Abstract

Heusler alloys constitute an interesting group of materials with wide applications. The purpose of the present study was to use the mechanical alloying method to synthesize Fe_2_CrSi Heusler alloy and learn about its structure and magnetic properties. Pure metal elements were ground for various periods of time in a planetary ball mill, and the process of alloy formation was monitored using X-ray diffraction and Mössbauer spectroscopy. It was found that after 20 h of milling, the disordered BCC solid solution was formed, with an average crystallite size ~11 nm. After thermal treatment, the desired Fe_2_CrSi Heusler alloy was obtained, with a small amount of secondary phases. Detailed XRD analysis showed the coexistence of two varieties of Heusler phase, namely Fm-3m and Pm-3n. The main result of this work is the detection of the hyperfine magnetic field distribution using Mössbauer spectroscopy. The occurrence of this distribution proves atomic disorder in the crystalline structure of the obtained Heusler alloy. Macroscopic magnetic measurements revealed soft magnetic properties of the alloy, with a magnetic moment of ~2.3 μ_B_/f.u., only slightly larger than the theoretically predicted value.

## 1. Introduction

A large family of Heusler alloys, i.e., full, inverse and half type, is the subject of both theoretical and experimental studies because of the interesting physical properties of these alloys. Namely, they possess various magnetic properties (i.e., ferromagnetism, ferrimagnetism, antiferromagnetism), as well as superconducting (e.g., in Pd_2_HfAl), semiconducting (in compounds with 24 valence electrons), giant magnetoresistive (in Co-based alloys), topological insulating (e.g., LuPtSb), magnetocaloric (e.g., Ni_2_MnGa), piezoelectric and catalytic properties. Owing to such a wealth of properties, materials of that type have a number of applications in spintronics, electronic and thermoelectric devices, superconductivity, tunnelling magnetoresistance, etc. [1,2,3,4], and as magnetic shape memory materials [5]. In the case of ternary compounds, full Heusler-type alloys are described by the formula X_2_YZ, where X and Y are transition metal elements and Z is an element from the III, IV or V group of the periodic table. Special attention is focused on Heusler alloys exhibiting half-metallic ferromagnetism (HMF), where the band structure of the ferromagnet is characterized as metallic for one spin state and simultaneously as semiconducting for another spin state. Half-metallic ferromagnets (HMFs) have a perfect (100%) spin-polarized current and can be used as spin injectors for magnetic random access memory devices [6].

An important issue is the methods of preparation of Heusler alloys. Classical arc-melting in an argon atmosphere and subsequent multiple re-melting for homogenization allows for obtaining alloys with good quality crystalline structure. Recently, it has been shown that a highly ordered structure of Heusler alloys may be obtained in the non-equilibrium process, i.e., novel Heusler single-phase alloy Fe_2_CrSi has been synthesized by means of melt-spinning [7]. However, except for the ordered phase, non-ordered or a mixture of ordered and non-ordered phases are often formed, which depends on the heat treatment. One of the non-equilibrium methods to produce materials, and also Heusler alloys, is mechanical alloying (MA). In general, the MA process consists of grinding the components of a given material/compound in a special mill. Recently, this method was used to obtain Fe-based full-Heusler alloys, i.e., Fe_2_CrSi and Co_2_FeAl [8,9,10]. Among them, Fe_2_CrSi is theoretically predicted to be HMF. The important issue is that during the MA process, Heusler alloys are formed as solid solutions, and only after annealing, the desired phase may be obtained. The preparation conditions, as well as the heat treatment, have a strong influence on the phase composition of the material.

In the present work, we tried to produce the Heusler alloy Fe_2_CrSi via the MA synthesis method in powder form. The novelty of our work is using Mössbauer spectroscopy to monitor the formation of the alloy at every stage of the milling process. Together with X-ray diffraction, Mössbauer spectroscopy allows us to recognize the phases and determine their magnetic state. The goal of the present paper was to study the influence of the annealing temperature on the phase composition and hyperfine interaction parameters of the obtained material. Moreover, macroscopic magnetic properties of the powders were determined.

## 2. Materials and Methods

A mixture of Fe, Cr and Si powders, corresponding to Fe_2_CrSi composition, was subjected to MA synthesis. The powder purity and particle size were better than 99.5% and below 100 μm, respectively. The milling process was performed in a planetary ball mill Fritsch Pulverisette P5 (Fritsch GmbH, Idar-Oberstein, Germany) operating at 200 rpm at room temperature. The vial and balls with a diameter of 10 mm were made of stainless steel. The total mass of powders was 10 g, and the ball-to-powder weight ratio was 10:1. The process of milling was conducted in an argon atmosphere. The material for further measurements was taken after 1, 2, 5, 10 and 20 h of MA synthesis. The samples after 20 h of MA were isothermally annealed for 1 h in an argon atmosphere at 300, 500 and 700 °C in a tube furnace.

The structure of ball-milled powders at every stage of MA synthesis was examined using X-ray diffraction (XRD). Measurements were carried out at room temperature on a Rigaku Mini Flex II (Rigaku, Tokyo, Japan) diffractometer using CuKα radiation (λ = 1.5418 Å). Phase analysis of the samples at different stages of the processing and after annealing was performed using PDF4+ database. The average size of the crystallites of the BCC solid solution formed after milling and Fe_2_CrSi compound obtained after heat treatment was estimated using the Scherrer formula. The lattice parameter was determined using Nelson–Riley method [11].

Differential Scanning Calorimetry (DSC, Perkin Elmer 8000, Perkin Elmer, Shelton, CT, USA) was applied to study the thermal stability of the structure obtained after milling. The samples were subjected to continuous heating from room temperature up to 700 °C at a heating rate of 20 °C min^−1^.

The local magnetism of ball-milled powders at every stage of MA synthesis was determined using ^57^Fe Mössbauer spectroscopy. The measurements were carried out using a conventional constant-acceleration-type spectrometer at room temperature with a ^57^Co (in a Rh matrix) as a source. The calibration of the spectrometer was carried out using α-Fe at room temperature.

The macroscopic magnetic properties of the materials were investigated with a vibrating sample magnetometer (VSM-VersaLab, Quantum Design, San Diego, CA, USA) under the magnetic field ranging from –2.0 T to +2.0 T at room temperature with an accuracy of ca. 0.1 °C. 

## 3. Results

### 3.1. Structural Studies

The synthesis process of Fe, Cr and Si elements in order to obtain Fe_2_CrSi Heusler alloy was monitored using X-ray diffraction (see Figure 1). The systematic disappearance of the Si peaks and overlapping of Fe and Cr diffraction lines indicate that a BCC solid solution was formed after 20 h of grinding. The average crystallite sizes, D, estimated from the broadening of the main diffraction line using the Scherrer formula for the final product of MA and the value of lattice parameter of the BCC solid solution, a, were as follows: D = 11 nm +/− 2 nm and a = 0.2873 nm +/− 0.0005 nm, respectively. The lattice parameter of the BCC solid solution is slightly higher than the lattice constant of pure iron (0.2866 nm) because silicon, having an atomic radius (110 pm) significantly smaller than that for iron (140 pm), forms an interstitial solid solution. This means that Si atoms are located in tetrahedral or octahedral sites existing in the BCC lattice and, thus, the lattice parameter of such a solution always increases, which we observe in our case. The obtained XRD results proved that the milling process did not lead to the formation of a Heusler Fe_2_CrSi phase. Milling for a longer period was not carried out because, as recently reported [8], grinding for a period of 40 h resulted in the formation of a α-(Fe, Cr, Si) BCC solid solution coupled with an amorphous phase.

In order to establish the suitable temperature for heat treatment, the DSC investigations were performed. Figure 2 presents the DSC curve obtained for the mixture of Fe, Cr and Si after 20 h of MA. Two broad exothermic peaks observed at 398 and 467 °C are presumably connected with the grain growth and subsequent solid-state reaction of mechanosynthesized powders via heat treatment. Because the later XRD results showed that after isothermal annealing at 500 °C, the Heusler alloy was formed, it can be concluded that the peak for T_p_ = 506 °C is related to the structural transformation from BCC solid solution to the Heusler phase.

Based on the DSC results and the literature data, three different annealing temperatures were chosen, i.e., 300, 500 and 700 °C. Figure 3 presents XRD patterns for the sample isothermally annealed at 300 °C together with the record for the 20 h milled powder. It can be noted that there are no changes between the XRD pattern for the BCC solid solution after 20 h of milling and for the sample subjected to annealing at 300 °C. 

However, the patterns for samples annealed at 500 and 700 °C show the separation of the diffraction peaks and the shift in the lines relative to the lines for the BCC solid solution (Figure 4). Additionally, the XRD pattern after heating the sample up to 700 °C in DSC is presented.

All diffraction lines in Figure 4 observed for the thermally treated powders can be assigned to two variants of the Heusler Fe_2_CrSi phase based on data from ICDD cards (no. 04-015-2526 and 04-005-1780). Both phases are cubic but differ slightly in the values of the lattice parameter, namely a = 0.5679 nm for the Fm-3m space group (black circles in Figure 4) and a = 0.4552 nm for the Pm-3n (black squares in Figure 4). The relative contribution of the phases was estimated as 86% of the Fm-3m phase and 14% of the Pm-3n phase, with neglection of the secondary phases. The lattice parameters of the Fe_2_CrSi alloy obtained after annealing at 500 and 700 °C are equal to a = 0.5666 nm +/− 0.0005 nm and a = 0.5668 nm +/− 0.0005 nm, respectively. Since the contribution of the Fm-3m phase is dominant, the lattice parameter of the obtained Heusler alloy is close to that for the pure Fm-3m phase, i.e., a = 0.5679 nm. In the pattern for the sample isothermally annealed at 700 °C, small amounts of the secondary phases were recognized, i.e., FeCr and Cr_3_Si. Thermal treatment led to grain expansion, and the average crystallite sizes were estimated as D = 18 nm +/− 4 nm and D = 30 nm +/− 5 nm for the 20 h milled powders subjected to isothermal annealing at 500 and 700 °C, respectively. It is worth noting that time is an important parameter in the heat treatment process. Figure 4 (blue line) shows that no secondary phases are visible in the XRD pattern for the sample heated continuously from room temperature up to 700 °C.

### 3.2. Mössbauer Spectroscopy Studies

Mössbauer spectra registered for the powders milled for 1, 2, 5, 10 and 20 h are presented in Figure 5. It can be seen that the spectra measured for the samples milled for 1 and 2 h have a single six-line pattern. As the time of milling increases, in the central part of the spectrum, an additional component appears, i.e., doublet.

From the numerical fitting of the spectra, hyperfine interaction parameters as well as the spectral parameters were obtained, and they are presented in Table 1. The observed sextet with the specific parameters shown in Table 1 is characteristic for iron. The iron component remains in all the spectra and has practically constant hyperfine interaction parameters; however, for the samples milled for 10 and 20 h, the width of spectral lines of the sextet slightly increased. This results from the fragmentation of grains to nanometer sizes. The doublet (red sub-spectrum in Figure 5b) may originate from the FeSi alloy because the values of isomer shift and quadrupole splitting determined for our samples are similar to those obtained for the Fe_50_Si_50_ melted alloy [12] as well as for Fe_50_Si_50_ nanopowders [13]. The relative contribution of the doublet to the whole Mössbauer spectrum is not large (~5% for 5 and 10 h and maximum 19% for the sample milled for 20 h). This is the reason that in XRD patterns, no clear lines from the FeSi phase are visible, but they may be hidden within the peaks coming from the BCC solid solution (Figure 4b, the main line of FeSi is around the 2 theta angle 45°). To obtain the best possible fitting parameter, the Mössbauer spectrum for 20 h MA was developed using doublet, iron sextet and the quasi-distribution of hyperfine magnetic fields (set of 6 green sextets in Figure 5b) with the assumption of the linear correlation between hyperfine magnetic field and isomer shift. The occurrence of such a distribution of components with broadened spectral lines and various hyperfine magnetic fields in the range of 9.07 and 26.86 T (see Table 1) proves that Cr and Si atoms are incorporated into the iron lattice, creating a BCC solid solution. 

After annealing the samples at 300, 500 and 700 °C, the registered Mössbauer spectra are visibly changed (Figure 6). The numerical fitting was performed using the paramagnetic component (single line and/or doublet) and the magnetic sextets. The obtained hyperfine interaction parameters of the components are shown in Table 2. The spectrum for the sample annealed at 300 °C consists of one doublet and three sextets, including one characteristic of iron. The presence of an iron component means that the desired Heusler Fe_2_CrSi phase has not been formed. This observation agrees well with the XRD results (in Figure 3, there are no visible differences between the patterns for 20 h MA and 300 °C). The doublet component may come from the FeSi phase, while the two sextets can be attributed to the BCC solid solution and/or the beginning of a newly formed Heusler phase. After annealing the samples at higher temperatures, the iron component in Mössbauer spectra disappears. The spectra for the samples annealed at 500 and 700 °C were numerically elaborated using one singlet, one doublet and the quasi-distribution of three or eight sextets, respectively. The single-line component, which has a relatively high contribution in the spectrum for the sample annealed at 700 °C, may be attributed to the FeCr secondary phase because its isomer shift has negative sign and the value similar to that reported for the alloy Fe_53.8_Cr_46.2_ [14] as well as for the FeCr nanoparticles [15]. The singlet with a contribution of 2.4% observed in the spectrum for the sample annealed at 500 °C also may come from the FeCr secondary phase, though there were no clear diffraction lines in the XRD pattern. However, Mössbauer spectroscopy is a much more sensitive technique than XRD; thus, it allows for the detection of very small amounts of phases. The doublet sub-spectrum fitted in the spectra for the samples annealed at 500 and 700 °C may originate from the FeSi secondary phase. 

The most important result from Mössbauer spectroscopy is the occurrence of the hyperfine magnetic field distribution detected for the samples annealed at 500 and 700 °C. In an ideal ordered L2_1_-type structure, which the Heusler Fe_2_CrSi possesses, the iron atoms are in 8c positions and have four Cr atoms and four Si atoms in the nearest neighborhood (Figure 6b). In this case, the Fe_2_CrSi phase has a small magnetic moment, and the Mössbauer spectrum should be a single sextet with a hyperfine magnetic field B_hf_ ~22 T [16]. However, in our experiment, a rather disordered Heusler Fe_2_CrSi alloy was obtained, which is proved by the broadened spectral lines (~0.3 mm/s). In addition to the 8c position in the Fm-3m phase, the iron atoms may occur at 4a or 4b sites, especially since the atomic radii of Fe, Cr and Si are very similar to each other. Moreover, as previously shown by XRD analysis, the Pm-3n phase also exists in the material; thus, the iron atoms can occupy the 6c or 2a positions. The possible atomic disorder causes the hyperfine magnetic field distribution observed in the Mössbauer spectra for the samples annealed at 500 and 700 °C.

### 3.3. Magnetic Measurements

The magnetization curves M(H) measured in a magnetic field up to 2 T for the powders milled for 1, 2, 5, 10 and 20 h show a course typical of the ferromagnetic material (Figure 7). The magnetization is relatively high and increases rapidly in a narrow range of weak fields below about 0.3 T and then tends to saturation in strong magnetic fields.

The values of the asymptotic saturation magnetization M_S_ were calculated using the relationship M (H) = M_S_(1 − b/H^2^), where b is the fitting parameter. Firstly, the M_S_(T) values slightly increase from 108.5 emu/g after 1 h of grinding to 119.5 emu/g after 2 and 5 h (Figure 8a). Next, the M_S_ decreases to about 98 emu/g after 20 h of MA. The magnetic moment μ per formal Fe_2_CrSi molecule was calculated according to the formula μ = (M_S_ × m/f.u.)/(N_A_ × μ_B_), where M_S_ is saturation magnetization, m—molecular mass per formula unit, N_A_—Avogadro number and μ_B_—Bohr magneton. For the powder milled for 1 h, the magnetic moment per formal Fe_2_CrSi molecule equals 3.72 μ_B_ and decreases to 3.36 μ_B_ after 20 h of grinding.

From the magnetic hysteresis loops, the coercive field H_C_ of the powders was determined and is presented in Figure 8b as a function of milling time. The sample milled for 1 h has a coercive field of H_C_ = 22 +/− 1 Oe. In powders ground from 2 to 20 h, H_C_ values increase monotonically from 31 +/− 1 to 59 +/− 1 Oe. Small values of the coercive field H_C_ and relatively high magnetization of the mechanically synthesized material confirm that all the tested samples belong to soft ferromagnets. It can be added that the magnetization of the powders decreases as the grinding time increases. This tendency is caused by the dilution of nonmagnetic Cr and Si elements in the iron lattice. Moreover, it is consistent with the core–shell model, because as the grain size decreases (down to 11 nm after 20 h of MA), the relative share of the structurally disordered surface layer increases [17]. After thermal treatment, the hysteresis loops are still narrow (Figure 9). 

As Mössbauer spectroscopy revealed, in the powder annealed at 300 °C the non-reacted iron exists, and this is the reason that the magnetization of the sample is only slightly smaller than that for the 20 h MA powder. After the annealing of powders at 500 and 700 °C, when the Fe_2_CrSi Heusler alloy was formed, the value of magnetization decreased significantly (to 66 emu/g for 500 °C and 68 emu/g for 700 °C). The coercive field was even smaller than that for 20 h milled powders, i.e., H_C_ = 10 +/− 1 Oe. The magnetic moment per formula unit equals 2.26 and 2.33 μ_B_ for the alloy obtained via annealing at 500 and 700 °C, respectively. The determined magnetic moments are larger than those predicted theoretically, i.e., 2 μ_B_/f.u. This result is justified because the theoretical magnetic moment of 2 μ_B_/f.u. is for an ideal Heusler phase with an ordered L2_1_ structure. However, our material after thermal treatment is, firstly, a mixture of two varieties of Heusler phases and, secondly, is characterized by atomic disorder, as shown by Mössbauer spectroscopy. 

## 4. Discussion

Indeed, the crystal structure and magnetic properties of the Fe_2_CrSi Heusler alloy prepared via the mechanical alloying method were studied and published recently by Lee [8] and Djaidi et al. [9]. The authors obtained the α-(Fe, Cr, Si) solid solution with a disordered BCC crystal structure after 20 h [8] or 24 h [9] of ball milling; however, in both cases, the milling process was conducted for longer times, i.e., 40 h [8] and 60 h [9]. In DSC curves, two exothermic peaks at 470 and 560 °C were observed for the powders milled for 40 h [8], while for the mixture milled for 32 h, three peaks at 452, 538 and 714 °C were registered [9]. In the case of our studies, three peaks at 398, 467 and 506 °C were detected. Such differences are understandable, because during the MA process, various materials of balls (tungsten carbide [8] or stainless steel [9] and our work) and vial (hardened steel [8] or steel [9]) were used; moreover, the ball-to-powder weight ratio was different, i.e., 7:1 [8], 17:1 [9] and 10:1 in our studies. After thermal treatment, in addition to the Fe_2_CrSi Heusler phase, the Cr_3_Si phase was recognized for the 40 h MA sample subjected to annealing at 650 °C [8] as well as for the 32 h milled powder and annealed at 600 °C [9]. In the present work, such a phase was registered for the 20 h MA sample isothermally annealed at 700 °C (small peaks marked by asterisks in Figure 4b); however, because this phase does not contain iron, it was not visible in the Mössbauer spectrum. Similar to Djaidi’s work, we observe two Bragg reflections at 2 theta angles ~27 and ~31°, which correspond to the superlattice reflections (111) and (200), respectively, characteristic of the Fe_2_CrSi Heusler phase (L2_1_).

It should be noted that long milling times, i.e., 40 or 60 h applied in the experiments by Lee and Djaidi to obtain Fe_2_CrSi Heusler alloy by mechanosynthesis, led to the formation of BCC solid solution together with an amorphous phase, which might be responsible for the low value of saturation magnetization (~30 emu/g [8,9]). Shorter milling times, in the order 20–30 h, and subsequent annealing at 500–600 °C allowed us to obtain the Heusler Fe_2_CrSi alloy with magnetization of about 55–80 emu/g and coercivity of 30–65 Oe [8,9]. Quite different values of the M_S_~64, 68, 59 emu/g and H_C_~157, 52, 41 Oe were reported for the Heusler nanoparticle alloy Fe_2_CrSi prepared using the mechanical alloying method in zirconia vial with zirconia balls during 4 h wet milling and subsequently annealed at 700, 800 and 900 °C, respectively [18]. In the case of our studies, the values of M_S_ agree well with the published data; however, the coercivity is relatively smaller. This proves that the synthesis conditions strongly influence the magnetic properties. One thing that can be stated is that the mechanically synthesized Fe_2_CrSi Heusler alloy obtained in our experiment possesses soft magnetic properties.

What contrasts our results from published data concerns the Heusler phase. As mentioned, after 20 h MA and isothermal annealing at 500 and 700 °C, the sample was the Fe_2_CrSi alloy, in which two Heusler variants coexist, i.e., Fm-3m and Pm-3n. This observation was possible by using two complementary methods, i.e., XRD and Mössbauer spectroscopy. Moreover, because of its high sensitivity, Mössbauer spectroscopy allowed us to detect small amounts of secondary phases containing iron. The coexistence of Fm-3m and Pm-3n Heusler variants may be responsible for the magnetic moment ~2.3 μ_B_/f.u. larger than that predicted theoretically.

## 5. Conclusions

It was shown that the Fe_2_CrSi Heusler alloy can be prepared by 20 h of mechanical alloying and subsequent thermal treatment. Both thermal processes, i.e., heating in the calorimeter (DSC) up to 700 °C and isothermal annealing in a tube furnace at 700 °C, resulted in the formation of the Heusler alloy, in which two Fm-3m and Pm-3n varieties coexist. Considering the XRD and Mössbauer spectroscopy results, it can be concluded that 20 h MA powder isothermally annealed at 700 °C in 80–90% constitutes the Fe_2_CrSi Heusler alloy with secondary FeSi, FeCr and Cr_3_Si phases. These silicon- and chromium-rich phases are paramagnetic, which may affect the macroscopic magnetization. It seems that slow heating up to 700 °C allows one to obtain the alloy without side phases. Both Mössbauer spectroscopy and magnetization measurements confirmed the ferromagnetic behavior of the samples. The main result of this work is the observation of the hyperfine magnetic field distribution in the samples annealed at 500 and 700 °C. The sextets with hyperfine magnetic fields B_hf_~20–22 T can be attributed to the iron atoms at 8c sites surrounded by four Cr atoms and four Si atoms (ordered L2_1_). However, the number of Cr and Si atoms in the nearest neighborhood of the Fe atom may be different in comparison with the ideal ordered L2_1_-type structure. Moreover, in addition to the 8c position, iron atoms may occur at 4a or 4b sites (in Fm-3m phase) or 6c or 2a positions (in Pm-3n phase). All these possibilities cause the atomic disorder revealed by the distribution of the hyperfine magnetic fields in Mössbauer spectra. 

## Figures and Tables

**Figure 1 nanomaterials-13-03024-f001:**
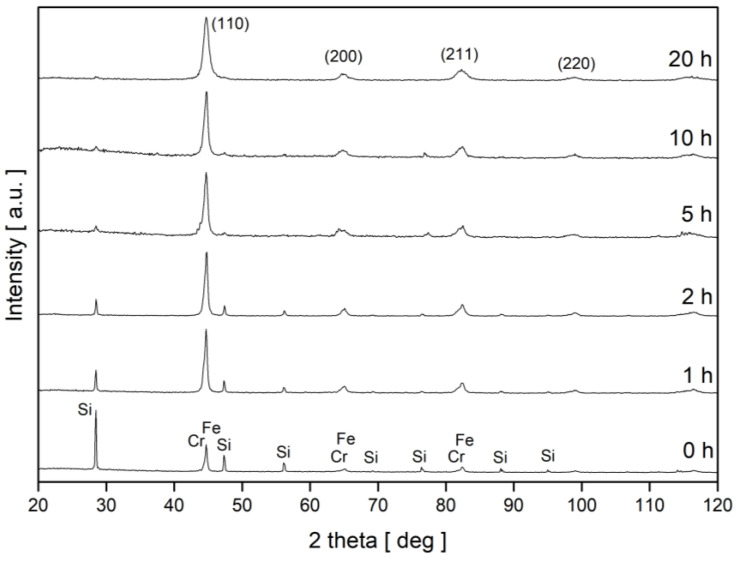
XRD patterns of a mixture of Fe, Cr and Si powders after various times of milling.

**Figure 2 nanomaterials-13-03024-f002:**
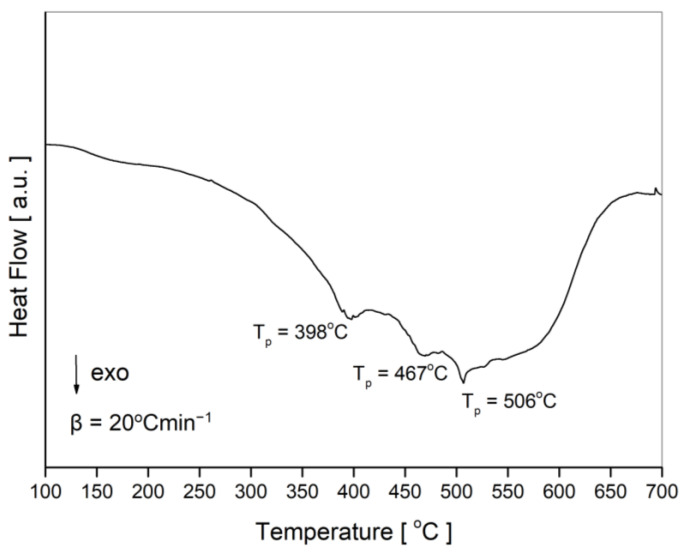
DSC curve for the BCC solid solution after 20 h of milling.

**Figure 3 nanomaterials-13-03024-f003:**
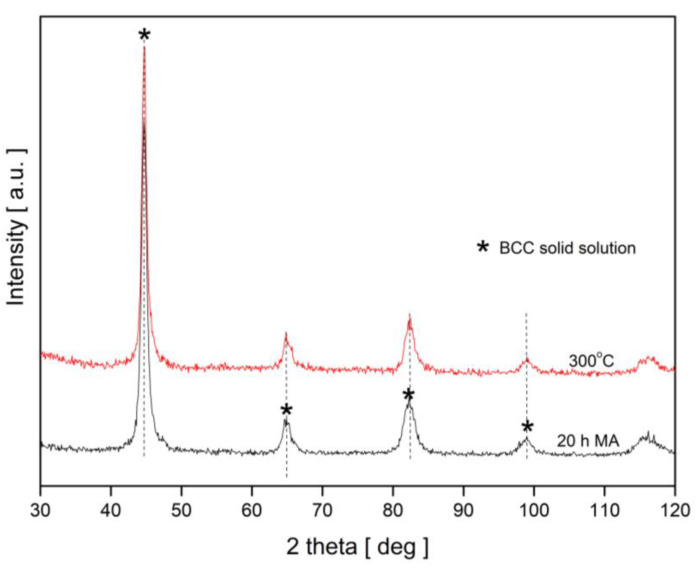
XRD patterns for BCC solid solution after 20 h of MA without and with subsequent isothermal annealing for 1 h at 300 °C.

**Figure 4 nanomaterials-13-03024-f004:**
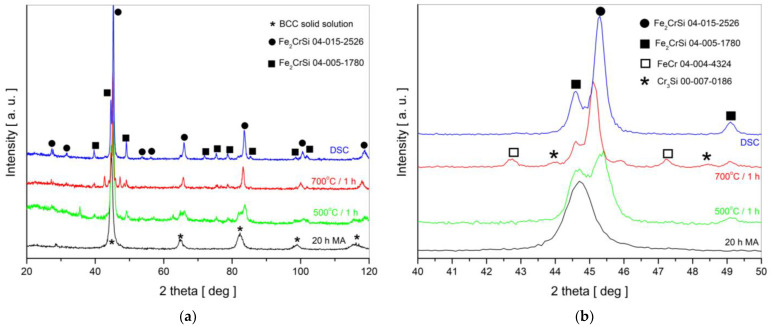
XRD patterns for: (**a**) 2 theta angle between 20 and 120 deg; (**b**) enlarged range of 2 theta angles around the main peak. Black line—pattern for the BCC solid solution after 20 h of MA; patterns for the samples after 20 h MA and isothermal annealing at 500 and 700 °C are marked in green and red, respectively; blue line—pattern for the BCC solid solution after 20 h of MA and heating in DSC up to 700 °C.

**Figure 5 nanomaterials-13-03024-f005:**
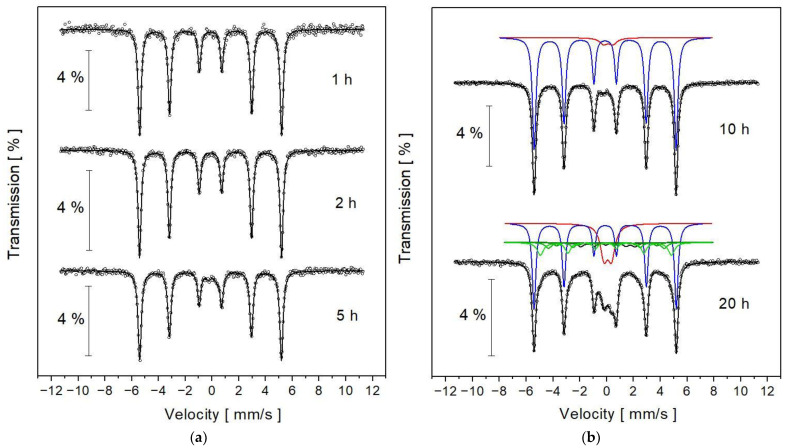
Room-temperature Mössbauer spectra for the mixture of Fe, Cr and Si powders subjected to ball milling during (**a**) 1, 2, 5 h and (**b**) 10 and 20 h. Discrete components marked for the samples milled for 10 and 20 h come from: iron—blue sextet, FeSi alloy—red doublet and BCC solid solution—set of 6 green sextets.

**Figure 6 nanomaterials-13-03024-f006:**
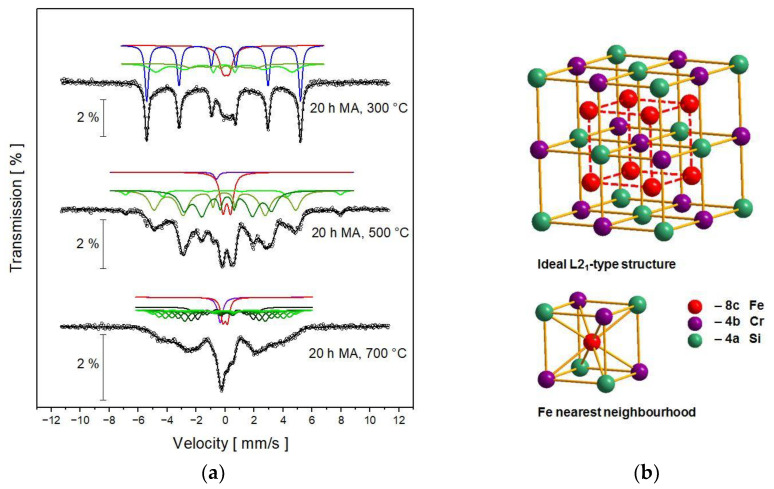
(**a**) Room-temperature Mössbauer spectra for the BCC solid solution after 20 h of MA and subsequent isothermal annealing for 1 h at 300, 500 and 700 °C; discrete components marked come from: iron—blue sextet, FeSi—red doublet, FeCr—purple singlet and Heusler Fe_2_CrSi alloy—set of green sextets; (**b**) the unit cell of the ideal L2_1_-type structure and the nearest neighbourhood of the iron atom.

**Figure 7 nanomaterials-13-03024-f007:**
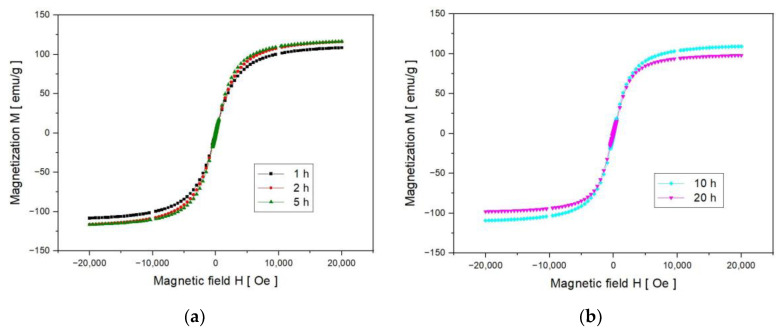
Magnetization curves for a mixture of Fe, Cr and Si powders after: (**a**) 1, 2 and 5 h and (**b**) 10 and 20 h of milling.

**Figure 8 nanomaterials-13-03024-f008:**
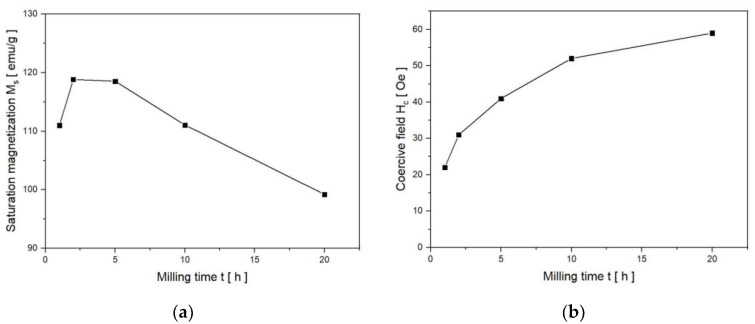
(**a**) Saturation magnetization determined with a relative accuracy of 1% and (**b**) coercive field (+/−1 Oe) as a function of milling time.

**Figure 9 nanomaterials-13-03024-f009:**
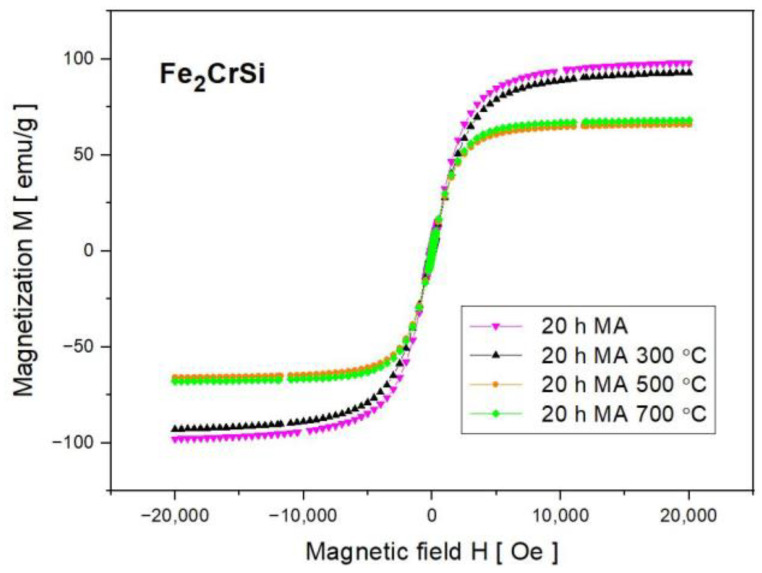
Magnetization for the samples after 20 h of MA without and with subsequent isothermal annealing for 1 h at 300, 500 and 700 °C.

**Table 1 nanomaterials-13-03024-t001:** Hyperfine interactions parameters derived from the discrete fit of the Mössbauer spectra for the powders milled for 1, 2, 5, 10 and 20 h; δ—isomer shift relative to α-Fe standard, B_hf_—hyperfine magnetic field, Δ—quadrupole splitting for the doublet, Γ—half width at half maximum of spectral lines, C—relative contribution of the component, χ^2^—fitting parameter; uncertainties are given in brackets for the last significant number.

Milling Time [h]	δ [mm/s]	B_hf_ [T]	Δ [mm/s]	Γ [mm/s]	C [%]	χ^2^	Component
1	−0.001 (1)	33.01 (1)	–	0.138 (2)	100	0.997	sextet α-Fe
2	−0.006 (1)	33.02 (1)	–	0.141 (2)	100	0.955	sextet α-Fe
5	0.158 (28)	–	0.622 (26)	0.257 (40)	5.4		doublet
	−0.011 (2)	32.96 (1)	–	0.146 (3)	94.6	0.808	sextet α-Fe
10	0.225 (20)	–	0.692 (19)	0.358 (35)	5.5		doublet
	−0.009 (3)	32.94 (1)	–	0.152 (2)	94.5	1.222	sextet α-Fe
20	0.183 (5)	–	0.556 (6)	0.325 (9)	18.9		doublet
	−0.006 (5)	33.01 (2)	–	0.153 (3)	51.5		sextet α-Fe
	0.323 (9)	9.07 (27)	–		4.3		sextet 1
	0.277 (9)	12.63 (27)	–		1.5		sextet 2
	0.231 (9)	16.19 (27)	–	0.240 (25) ^1^	1.0	1.005	sextet 3
	0.185 (9)	19.75 (27)	–		3.4		sextet 4
	0.139 (9)	23.31 (27)	–		6.0		sextet 5
	0.093 (9)	26.86 (27)	–		13.4		sextet 6

^1^ All 6 sextets have the same width of spectral lines.

**Table 2 nanomaterials-13-03024-t002:** Hyperfine interactions parameters derived from the discrete fit of the Mössbauer spectra for the powders milled for 20 h and annealed at 300, 500 and 700 °C, respectively; δ—isomer shift relative to α-Fe standard, B_hf_—hyperfine magnetic field, Δ—quadrupole splitting for the doublet, Γ—half width at half maximum of spectral lines, C—relative contribution of the component, χ^2^—fitting parameter; uncertainties are given in brackets for the last significant number.

T [°C]	δ [mm/s]	B_hf_ [T]	Δ [mm/s]	Γ [mm/s]	C [%]	χ^2^	Component
300	0.172 (6)	–	0.418 (8)	0.317 (14)	19.3		doublet
	0.007 (2)	33.03 (1)	–	0.137 (5)	47.1	1.162	sextet α-Fe
	0.028 (11)	29.24 (18)	–	0.395 (45)	23.1		sextet 1
	0.054 (25)	14.11 (41)	–	0.395 (45)	0.5		sextet 2
500	−0.500 (24)	–	–	0.183 (39)	2.4		singlet
	0.214 (5)	–	0.530 (5)	0.230 (6)	15.8		doublet
	0.368 (15)	46.20 (12)	–		5.6	2.362	sextet 1
	0.056 (5)	30.44 (6)	–	0.334 (16) ^1^	38.3		sextet 2
	0.265 (7)	18.77 (5)	–		37.9		sextet 3
700	−0.238 (10)	–	–	0.209 (14)	13.6		singlet
	0.107 (11)	–	0.320 (7)	0.176 (12)	9.8		doublet
	0.217 (4)	9.28 (14)	–		13.6		sextet 1
	0.191 (4)	11.95 (14)	–		11.5		sextet 2
	0.163 (4)	14.62 (14)	–		12.4	0.834	sextet 3
	0.136 (4)	17.29 (14)	–	0.241 (15) ^2^	9.2		sextet 4
	0.109 (4)	19.96 (14)	–		8.4		sextet 5
	0.082 (4)	22.64 (14)	–		9.0		sextet 6
	0.055 (4)	25.31 (14)	–		7.9		sextet 7
	0.028 (4)	27.97 (14)	–		4.4		sextet 8

^1^ All 3 sextets have the same width of spectral lines. ^2^ All 8 sextets have the same width of spectral lines.

## Data Availability

Data are contained within the article.

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
