# Peer review of "Structure and Magnetic Properties of Mechanosynthesized Nanocrystalline Fe2CrSi Heusler Alloy"

_nanomaterials, 2023, doi:10.3390/nano13233024_

Round 1

Reviewer 1 Report

Comments and Suggestions for Authors

In this manuscript, Fe2CrSi Heusler alloy was prepared with mechanical alloying method. The structure and magnetic properties were then investigated. The authors found the coexistence of Heusler phase Fm-3m and Pm-3n in the sample after thermal treatment. Based on Mössbauer spectroscopy, hyperfine magnetic field distribution was detected which proves the atomic disorder in the crystalline structure of the obtained Heusler alloy. Besides, the authors also revelaed soft magnetic properties of the alloy which is accordance with the theoretical results. The manuscript is well organized, technically well written. However, major revisions need to be made concerning the following issues before acceptance.

1. The authors stated that “The peak at Tp = 506 °C can be connected with the phase transformation…”. However, peaks observed at 398 and 467 °C were connected with the grain growth. Please explain. Besides, after 506 °C, the heat flow increase with the increasing temperature. What is the cause of this phenomenon?

2. The authors stated that “This results from the fragmentation of grains to nanometer size…”. How was the nanometer size of the sample determined ? Perhaps SEM or TEM results are needed.

3. The magnetization curves measured in a magnetic field up to 2 T for the powders 235 milled were shown in Figure 7. Why the curves for for 1, 2, 5 are seperated with that of 10 and 20 h? The authors should explain more on the relationship between magnetic properties with the milling time.

4. The authors stated that “The magnetic moment per formal Fe2CrSi molecule equals to 3.84 μB for the powder milled for 1 h, and decreases to 3.47 μB after 20 h of grinding…”. However, no supporting information can be found for this statement. The manuscript does not give a formula for calculating the magnetic moment or a method for fitting the magnetic moment.

5. The authors stated that “The determined magnetic moments are larger than that predicted theoretically…”. What is the reason that the experimental magnetic moment is larger than the theoretical value? Please explain.

Author Response

We would like to thank the Reviewer for detailed remarks, which allowed us to improve the quality of our manuscript. All corrections in the text of the work are marked in red. We hope that the answers to the questions will satisfy the Reviewer and the revised manuscript will be approved for publication in Nanomaterials.

  1. The authors stated that “The peak at Tp = 506 °C can be connected with the phase transformation…”. However, peaks observed at 398 and 467 °C were connected with the grain growth. Please explain. Besides, after 506 °C, the heat flow increase with the increasing temperature. What is the cause of this phenomenon?

Answer: Because the later XRD results showed that after isothermal annealing at 500 °C, the Heusler alloy was formed, it can be concluded that the peak for Tp = 506 °C is related to the structural transformation from BCC solid solution to the Heusler phase. The broadened peaks observed at 398 and 467 °C can be connected both with the grain growth and the successively emerging Heusler phase.

  1. The authors stated that “This results from the fragmentation of grains to nanometer size…”. How was the nanometer size of the sample determined ? Perhaps SEM or TEM results are needed.

Answer: The average crystallite sizes, D, were estimated from the broadening of the main diffraction line using the Scherrer formula. The information about this was included in the original version of the work in paragraph 2, line 78 and in paragraph 3.1, line 96. In the present version it is in lines 84 and 103.

  1. The magnetization curves measured in a magnetic field up to 2 T for the powders 235 milled were shown in Figure 7. Why the curves for for 1, 2, 5 are seperated with that of 10 and 20 h? The authors should explain more on the relationship between magnetic properties with the milling time.

Answer: In Figure 7, the curves for 1, 2, 5 h are separately drawn from that of 10 and 20 h for better readability of the drawing, because all hysteresis loops would run into each other. The magnetization of the powders decreases as the grinding time increases. This tendency is caused by the dilution of nonmagnetic Cr and Si elements in the iron lattice. Moreover, it is consistent with the core–shell model, because as the grain size decreases (down to 11 nm after 20 h of MA), the relative share of the structurally disordered surface layer increases. This explanation was made in the original version of the manuscript (lines 255-259), in the present version in lines 285-290.

  1. The authors stated that “The magnetic moment per formal Fe2CrSi molecule equals to 3.84 μfor the powder milled for 1 h, and decreases to 3.47 μafter 20 h of grinding…”. However, no supporting information can be found for this statement. The manuscript does not give a formula for calculating the magnetic moment or a method for fitting the magnetic moment.

Answer: The magnetic moment m per formal Fe2CrSi molecule was calculated according to the formula m = (MS · m/f.u.)/(NA · mB), where MS is saturation magnetization, m – molecular mass per formula unit, NA – Avogadro number and mB Bohr magneton. By the way, a mistake was discovered in the calculations, i.e., error in molecular mass, thus the present values of magnetic moments per formula unit have smaller values in comparison with the previous version of our manuscript.

  1. The authors stated that “The determined magnetic moments are larger than that predicted theoretically…”. What is the reason that the experimental magnetic moment is larger than the theoretical value? Please explain.

Answer: The determined magnetic moments are larger than that predicted theoretically, i.e., 2 mB/f.u.. This result is justified because theoretical magnetic moment of 2 mB/f.u. is for an ideal Heusler phase with an ordered L21 structure. However, our material after thermal treatment is, firstly, a mixture of two varieties of Heusler phases, and secondly is characterized by atomic disorder, as shown by Mössbauer spectroscopy.

Reviewer 2 Report

Comments and Suggestions for Authors

1.      Line 27 “A large family of Heusler alloys, i.e., full, inverse and half type, is a subject of both 27 theoretical and experimental studies because of interesting physical properties….”

Their interesting physical properties should be listed.

2.      Line 98 “The lattice parameter of the BCC solid solution is slightly higher from the lattice constant of pure iron.”

The authors are asked to formulate an explanation regarding this finding.

3.      Line 110 “The peak at Tp = 506 0C can be connected with the phase transformation”

Authors should specify which phase transformation is involved.

4.      Line 127 “The detailed analysis of XRD results was performed for the samples annealed at 500 and 700 0C.”

What does detailed analysis mean? How does it differ from previous analyses?

5.      Line 128 “Figure 4 presents the comparison of the XRD patterns for the BCC solid solution after 20 h of milling and for the samples after isothermal annealing at 500 and 700 0C.”

Figures 3 and 4 show the same, with the difference that the DSC sample spectra have been added to Figure 4. This could be added from the start to Figure 3, so I don't see the point of Figure 4!

6.      Line 130 “Additionally, XRD pattern after heating the sample up to 700 0C in DSC is presented.”

It is presented but not discussed. Although, it seems that this sample does not present secondary phases! The annealing time seems to be an important parameter that the authors did not consider.

7.      Details of how the magnetic moment was calculated should be provided.

 8.      Line 250 “From the magnetic hysteresis loops measured in a magnetic field from -0.03 to 0.03 T the coercive field HC of the powders were determined (Figure 8b).”

A hysteresis curve measured in a magnetic field from -0.03 to 0.03 T is a minor curve. The determination of Hc from minor hysteresis curves is not relevant.

9.      Line 270 “The determined magnetic moments are larger than that predicted theoretically, i.e., 2 μB/f.u.”

How do you comment on this finding?

10.  Line 294 “Comparing the macroscopic magnetic properties of mechanosynthesized Fe2CrSi Heusler alloy……….”

It must be reformulated. It is not clear with whom it is compared

11.  I appreciate the comparison with the data already published by other authors, but don't the authors emphasize the necessity of their study? The authors should state what is the added value of this study in the field and why it is so important.

Why is it so important that two-phase variants coexist, Fm-3m and Pm-3n? What is the gain in relation to the magnetic properties?

Comments on the Quality of English Language

Some phrases like Comparing the macroscopic magnetic properties of mechanosynthesized Fe2CrSi Heusler alloy……….” should be phrased for full understanding.

Author Response

We would like to thank the Reviewer for detailed remarks, which allowed us to improve the quality of our manuscript. All corrections in the text of the work are marked in red. We hope that the answers to the questions will satisfy the Reviewer and the revised manuscript will be approved for publication in Nanomaterials.

  1. Line 27 “A large family of Heusler alloys, i.e., full, inverse and half type, is a subject of both theoretical and experimental studies because of interesting physical properties….”

Their interesting physical properties should be listed.

Answer: the list of properties was added in the second sentence of Introduction.

  1. Line 98 “The lattice parameter of the BCC solid solution is slightly higher from the lattice constant of pure iron.”

The authors are asked to formulate an explanation regarding this finding.

Answer: When a solid solution is created in the Fe matrix with a BCC lattice, Cr atoms replace Fe atoms in the lattice sites. It is a substitutional solid solution. In such a case, the lattice parameter of the solid solution may increase or decrease, depending on the size of the atomic radius of the dissolved element. Here, the atomic radii of Fe (140 pm) and Cr (140 pm) are the same. However silicon, having an atomic radius (110 pm) significantly smaller than that for iron (140 pm), forms an interstitial solid solution. This means that Si atoms are located in tetrahedral or octahedral sites existing in the BCC lattice and thus the lattice parameter of such a solution always increases, which we observe in our case.

  1. Line 110 “The peak at Tp = 506 0C can be connected with the phase transformation”

Authors should specify which phase transformation is involved.

Answer: Because the later XRD results showed that after isothermal annealing at 500 °C, the Heusler alloy was formed, it can be concluded that the peak for Tp = 506 °C is related to the structural transformation from BCC solid solution to the Heusler phase.

  1. Line 127 “The detailed analysis of XRD results was performed for the samples annealed at 500 and 700 0C.”

What does detailed analysis mean? How does it differ from previous analyses?

Answer: This sentence was deleted.

  1. Line 128 “Figure 4 presents the comparison of the XRD patterns for the BCC solid solution after 20 h of milling and for the samples after isothermal annealing at 500 and 700 0C.”

Figures 3 and 4 show the same, with the difference that the DSC sample spectra have been added to Figure 4. This could be added from the start to Figure 3, so I don't see the point of Figure 4!

Answer: In Figure 3 only XRD patterns for 20 h MA and the sample annealed for 300 °C were presented. The description of the XRD results was corrected.

  1. Line 130 “Additionally, XRD pattern after heating the sample up to 700 0C in DSC is presented.”

It is presented but not discussed. Although, it seems that this sample does not present secondary phases! The annealing time seems to be an important parameter that the authors did not consider.

Answer: The following comment was added at the end of the 3.1. paragraph:

It is worth noting that time is an important parameter in the heat treatment process. Figure 4 (blue line) shows that no secondary phases are visible in the XRD pattern for the sample heated continuously from room temperature up to 700 °C.

  1. Details of how the magnetic moment was calculated should be provided.

Answer: The magnetic moment m per formal Fe2CrSi molecule was calculated according to the formula m = (MS · m/f.u.)/(NA · mB), where MS is saturation magnetization, m – molecular mass per formula unit, NA – Avogadro number and mB Bohr magneton. By the way, a mistake was discovered in the calculations, i.e., error in molecular mass, thus the present values of magnetic moments per formula unit have smaller values in comparison with the previous version of our manuscript.

  1. Line 250 “From the magnetic hysteresis loops measured in a magnetic field from -0.03 to 0.03 T the coercive field HC of the powders were determined (Figure 8b).”

A hysteresis curve measured in a magnetic field from -0.03 to 0.03 T is a minor curve. The determination of Hc from minor hysteresis curves is not relevant.

Answer: The coercive field HC was determined from the magnetic hysteresis loops which were measured separately in a magnetic field from -0.03 to 0.03 T. The values of HC are given with the accuracy of +/- 1 Oe.

  1. Line 270 “The determined magnetic moments are larger than that predicted theoretically, i.e., 2 μB/f.u.” How do you comment on this finding?

Answer: The determined magnetic moments are larger than that predicted theoretically, i.e., 2 mB/f.u.. This result is justified because theoretical magnetic moment of 2 mB/f.u. is for an ideal Heusler phase with an ordered L21 structure. However, our material after thermal treatment is, firstly, a mixture of two varieties of Heusler phases, and secondly is characterized by atomic disorder, as shown by Mössbauer spectroscopy.

  1. Line 294 “Comparing the macroscopic magnetic properties of mechanosynthesized Fe2CrSi Heusler alloy……….”

It must be reformulated. It is not clear with whom it is compared

Answer: The sentence was reformulated to be more clear.

  1. I appreciate the comparison with the data already published by other authors, but don't the authors emphasize the necessity of their study? The authors should state what is the added value of this study in the field and why it is so important. Why is it so important that two-phase variants coexist, Fm-3m and Pm-3n? What is the gain in relation to the magnetic properties?

Answer: The novelty of our work is using the Mössbauer spectroscopy to monitor the formation of the alloy at every stage of milling process. Together with X-ray diffraction, Mössbauer spectroscopy allows us to recognize the phases and determine their magnetic state. In the published works, the authors did not use Mössbauer spectroscopy. We added one more reference, i.e., [18], to prove that the synthesis conditions strongly influence the magnetic properties. Thus, the coexistence of Fm-3m and Pm-3n Heusler variants discovered in our studies can be responsible for the magnetic moment larger than that predicted theoretically.

Comments on the Quality of English Language

Some phrases like “Comparing the macroscopic magnetic properties of mechanosynthesized Fe2CrSi Heusler alloy……….” should be phrased for full understanding.

Answer: The sentence was reformulated to be more clear.

Reviewer 3 Report

Comments and Suggestions for Authors

The authors present the structure and magnetic properties of Fe2CrSi synthesized by MA technique. This technique for  Fe2CrSi is well discussed in Ref.8-10. As written in lines 55-59 the novelty of this article is the Mossbauer spectroscopy studies reported here.

There are several points which need clarification.

Line 149. How  the grain sizes was determined?

Lines 246-247. How were calculated the magnetic moments?

No error bars in Fig. 8a,b  and no uncertainties in Hc (Lines 252-253).

My major concern is the data and the provided interpretation   in Tables 1-2.

Table 1.

The line width of pure Fe is the same for all six lines.   It is not physically correct to use a different line widths for the same material.

Why is the effect size for 5h is 2% ?

For 10 h. The broad line width (0.36 mm/s) clearly indicates a distribution in quadrupole splitting.

Additional 6 sextets are displayed. They differ in their IS and Heff values. Why only 6 sextets? What is the meaning of a sextet with C=0.9%. What are the line-widths of sextets 1,2 and 6?.

It seems that for h=20 h spectrum, the central part can be fitted with one doublet and one sextet, in which the quadrupole splitting and the effective moments are distributed. Six sextets with different IS and B values, are meaningless.

The quadrupole shift column has no meaning and can be omitted.

Table 2.

T=300 C. Two sextets with 3 line-widths. It is not clear.

T=500C. Very high Heff =46.2 kOe, line-widths, and fitting parameters. The spectrum needs a better analysis.

The spectrum must be re-analyzed without the singlet.

Author Response

We would like to thank the Reviewer for detailed remarks, which allowed us to improve the quality of our manuscript. All corrections in the text of the work are marked in red. We hope that the answers to the questions will satisfy the Reviewer and the revised manuscript will be approved for publication in Nanomaterials.

  1. Line 149. How  the grain sizes was determined?

Answer: The average crystallite sizes, D, were estimated from the broadening of the main diffraction line using the Scherrer formula. The information about this was included in the original version of the work in paragraph 2, line 78 and in paragraph 3.1, line 96. In the present version it is in lines 84 and 103.

  1. Lines 246-247. How were calculated the magnetic moments?

Answer: The magnetic moment m per formal Fe2CrSi molecule was calculated according to the formula m = (MS · m/f.u.)/(NA · mB), where MS is saturation magnetization, m – molecular mass per formula unit, NA – Avogadro number and mB Bohr magneton. By the way, a mistake was discovered in the calculations, i.e. error in molecular mass, thus the present values of magnetic moments per formula unit have smaller values in comparison with the previous version of our manuscript.

  1. No error bars in Fig. 8a,b  and no uncertainties in Hc (Lines 252-253).

Answer: The saturation magnetization was determined with a relative accuracy of 1 % and coercive field with +/- 1 Oe. The error bars would be too small to mark in the drawing. The information concerning accuracy is in the caption of Figure 8 and in the text.

  1. Table 1. The line width of pure Fe is the same for all six lines.   It is not physically correct to use a different line widths for the same material.

Answer: The line widths for spectral lines were veryfied and corrected in Table 1. The lines in sextets in quasi-distributions have the same widths, what was marked in footer to Table 1.

  1. Why is the effect size for 5h is 2% ?

Answer: We apologize for this error, it occurred while preparing the drawings, and the spectra for the samples after MA synthesis were confused with those after annealing, where the effect size was 2%.

  1. For 10 h. The broad line width (0.36 mm/s) clearly indicates a distribution in quadrupole splitting.

Answer: The doublet with broad line width (red sub-spectrum in Figure 5b) may origin from the FeSi alloy which has fragmented and deformed structure. In XRD patterns no clear lines from FeSi phase are visible, but they may be hidden within the peaks coming from the BCC solid solution (Figure 4b, main line of FeSi is around the 2 theta angle 45°).

  1. Additional 6 sextets are displayed. They differ in their IS and Heff values. Why only 6 sextets? What is the meaning of a sextet with C=0.9%. What are the line-widths of sextets 1,2 and 6?.

Answer: The spectra were fitted by the discrete model, thus the using of 6 sextets should be named quasi-distribution. It was added in the text. The estimated C values were rounded differently. The line widths were corrected.

  1. It seems that for h=20 h spectrum, the central part can be fitted with one doublet and one sextet, in which the quadrupole splitting and the effective moments are distributed. Six sextets with different IS and B values, are meaningless.

Answer: In the fitting procedure the quasi-distribution was used and we consider that various hyperfine magnetic fields testifies different nearest neighbourhood of iron in the BCC solid solution.

  1. The quadrupole shift column has no meaning and can be omitted.

Answer: The values of quadrupole shift for the sextets were deleted from Tables 1 and 2.

  1. Table 2. T=300 C. Two sextets with 3 line-widths. It is not clear.

Answer: The line widths for spectral lines were veryfied and corrected in Table 2. The lines in sextets in quasi-distributions have the same widths, what was marked in footer to Table 2.

  1. T=500C. Very high Heff =46.2 kOe, line-widths, and fitting parameters. The spectrum needs a better analysis. The spectrum must be re-analyzed without the singlet.

Answer:  By the way, an error in the sign of the value of isomer shift was detected (it was corrected in Table 2). The singlet with contribution 2.4 % observed in the spectrum for the sample annealed at 500 °C also may come from FeCr secondary phase though there were no clear diffraction lines in XRD pattern. However, Mössbauer spectroscopy is a much more sensitive technique than XRD, thus it allows the detection of very small amounts of phases. For the sextet with high hyperfine field 46.2 T we have no explanation.

Round 2

Reviewer 1 Report

Comments and Suggestions for Authors

I recommend to accept the paper for publication since the comments of the reviewers have been properly answered.

Author Response

Thank you very much for your positive opinion about our manuscript.

Reviewer 2 Report

Comments and Suggestions for Authors

The coercive field is determined from the major hysteresis curve, the one already measured between -2 and +2 T. There is no need for another separate measurement between -0.03 and 0.03 T. This is, as I told you before, a curve of minor hysteresis, and the Hc determined in this case may differ from the Hc value determined from the major curve.

I suggest you only replace the value of Hc determined from the minor curve with the value determined from the major curve and delete the expression "From the magnetic hysteresis loops measured separately in a magnetic field from -0.03 to 0.03 T the coercive field HC of the powders were determined (Figure 8b)."

 Without specifying anything, it will be understood by itself that Hc is determined from the hysteresis curve.

Author Response

Reviewer: I suggest you only replace the value of Hc determined from the minor curve with the value determined from the major curve and delete the expression "From the magnetic hysteresis loops measured separately in a magnetic field from -0.03 to 0.03 T the coercive field HC of the powders were determined (Figure 8b).

Answer: Thank you for this remark. We have verified the values of Hc from the major hysteresis loops and they are practically the same as presented in Figure 8b in the limit of the experimental error.

The sentence: „From the magnetic hysteresis loops measured separately in a magnetic field from -0.03 to 0.03 T the coercive field HC of the powders were determined (Figure 8).” was deleted and changed for:

„From the magnetic hysteresis loops, the coercive field HC of the powders were determined and presented in Figure 8b as a function of milling time.”  Changes are highlighted in yellow in lines 281-285.

Reviewer 3 Report

Comments and Suggestions for Authors

I am pleased with the corrections made.

Author Response

(The authors gave the same response as above.)
